# Spectrum of Clinical Features and Genetic Profile of Left Ventricular Noncompaction Cardiomyopathy in Children

**Agata Paszkowska [1], Alicja Mirecka-Rola [1], Dorota Piekutowska-Abramczuk [2], Elżbieta Ciara [2]**, **Łukasz Mazurkiewicz [3], Katarzyna Bieganowska [1] and Lidia Ziółkowska [1,*]**

1   Department of Cardiology, The Children's Memorial Health Institute, 04-730 Warsaw, Poland;
    a.paszkowska@ipczd.pl (A.P.); a.mirecka-rola@ipczd.pl (A.M.-R.); k.bieganowska@ipczd.pl (K.B.)
2   Department of Medical Genetics, The Children's Memorial Health Institute, 04-730 Warsaw, Poland;
    D.Abramczuk@IPCZD.PL (D.P.-A.); e.ciara@ipczd.pl (E.C.)
3   Departmentof Cardiomyopathies, Cardiovascular Magnetic Resonance Unit, National Institute of Cardiology,
    04-682 Warsaw, Poland; lmazurkiewicz@ikard.pl
*   Correspondence: l.ziolkowska@ipczd.pl; Tel.: +48-22-8157370

**Abstract:** Background: Left ventricular noncompaction (LVNC) is a genetically determined car-
diomyopathy that occurs following a disruption of endomyocardial morphogenesis. The purpose
of this study was to identify the clinical characteristics and genetic profile of children with LVNC.
Methods: From February 2008 to July 2020, a total of 32 children (median 11.5 years) with LVNC were
prospectively enrolled and followed up for a median of 4.02 years. Diagnosis was made based on
characteristic features of LVNC in echocardiography and cardiovascular magnetic resonance (CMR).
Patients' clinical symptoms, family history, ECG, Holter ECG, and genetic tests were also evaluated.
Results: The most common presenting symptom was heart failure (31% of children). ECG abnormali-
ties were noted in 56% of patients. The most prominent features were ventricular arrhythmias, sinus
bradycardia, and paroxysmal third-degree atrioventricular block. Most of the patients (94%) met the
criteria for LVNC and CMR confirmed this diagnosis in 82% of cases. The molecular etiology was
found in 53% of children. Conclusion: Although heart failure and arrhythmias were very frequent in
our study group, thromboembolic events and genetic syndromes were rare. For the accurate and
reliable assessment of children with LVNC, it is necessary to get to know their family history and
detailed clinical profile.

**Keywords:** left ventricular noncompaction; cardiomyopathy; heart failure; arrhythmia; conduction
disturbances; molecular etiology; children

## 1. Introduction

Left ventricular noncompaction cardiomyopathy (LVNC) is a genetically determined myocardial disease, the third most common cardiomyopathy in the pediatric population (after dilated and hypertrophic cardiomyopathy) [1]. Molecular studies confirm the genetic basis in approximately 40% of LVNC patients [2]. The clinical presentation of LVNC varies widely, ranging from asymptomatic cases to severe heart failure, arrhythmias, thromboembolic complications, and sudden cardiac death [3,4]. Heart failure is the most common clinical symptom, occurring in approximately 55% of patients with LVNC [5]. The spectrum of arrhythmias is very wide, with the most common one being ventricular arrhythmia. Left ventricular (LV) systolic dysfunction, arrhythmias, and blood stasis in the recesses of the myocardium predispose to thromboembolic events [6–8]. The basic diagnostic examination in LVNC is echocardiography. While LVNC is generally diagnosed based on published echocardiographic criteria [9], cardiovascular magnetic resonance (CMR) can also be useful in suspected cases of LVNC and may help determine the prognosis by detecting fibrosis [10,11]. Therapeutic management includes the treatment of heart failure, cardiac arrhythmias and thromboembolic complications.

This study sought to determine the clinical features and genetic causes of LVNC in children diagnosed with this disease at a single institution.

## 2. What Is Missing?

An assessment of clinical presentation in a homogeneous group of pediatric patients with an isolated form of LVNC has not, to date, been conducted.

Furthermore, there are still no uniform recommendations for the prevention of thromboembolic events in children with LVNC.

So far, no diagnostic criteria in imaging studies developed for the pediatric population with LVNC have been established. Is echocardiography sufficient to diagnose childhood LVNC?

## 3. Materials and Methods

**Study patients**. From March 2008 to July 2020 pediatric patients with diagnosed LVNC hospitalized in the Department of Cardiology of the Children's Memorial Health Institute were prospectively enrolled. The criteria for inclusion in the study were age < 18 years at the time of diagnosis and echocardiographic evidence of isolated LVNC defined as: 1. The presence of a two-layer structure with a compacted (C) and noncompacted (NC) endocardial layer of the trabecular meshwork with deep endomyocardial spaces. 2. A maximal end-systolic ratio of NC/C layers of >2.3. Color Doppler evidence of deep perfused intertrabecular recesses [9].

The Institutional Ethics Committee approved this study. Informed consent was obtained from all individual participants included in the study.

**Data collection**. Patients' demographics, clinical symptoms, family history of cardiomyopathies and sudden cardiac death (SCD), arrhythmias in family members, and treatment strategy, as well as the echocardiography, 12-lead resting ECG, 24-h Holter ECG, cardiopulmonary exercise test (CPET), and CMR results, were collected. In all children, the NYHA/Ross functional class and clinical symptoms such as chest pain, palpitations, syncope, pre-syncope, and thromboembolic events were evaluated. Serum NT-proBNP levels were assessed in all patients. Each patient underwent genetic blood tests for evaluation of the molecular basis of the disease. DNA from the peripheral blood was extracted automatically using the MagCore Nucleic acid Extractor HF16Plus. Next-generation sequencing (targeted panel of 164 cardiomyopathy-associated genes) was applied in all cases. The sequencing was performed on the HiSeq 1500 platform (Illumina). The discovered variants were analyzed and prioritized considering: 1. The minor allele frequency determined with the gnomAD, ExAC, 10UK, and in-house databases, containing data from more than 5000 individuals. 2. The pathogenicity of the variants, using up to 13 different in silico prediction algorithms (CADD, SIFT, MutationTaster, PolyPhen2 (HDIV, HVAR), MutationAssessor, LRT, MetaSVM, MetaLR, FATHMM, MaxEnt, NNSPLICE and SSF). 3. Phenotypic descriptions in OMIM, GTR, HGMD, ClinVar, Varsome, and Pubmed. Sanger sequencing was used for validation of the most interesting candidate variants and for parental segregation analysis when available.

Two-dimensional, Doppler, and M-mode echocardiography were performed at rest using standard methods. Echocardiographic measurements included the LV end-diastolic dimension, the left ventricular ejection fraction (LVEF) (measured according to the Simpson method), and LV noncompaction features (measured via the calculation of the NC/C ratio and visualization of the recess filling between the trabeculae with blood flowing in from LV using the color Doppler method as reported in the literature) [9,12]. The 24-h Holter monitors and 12-lead resting ECGs were also examined for evidence of supraventricular and ventricular arrhythmia, sinus node disease, sinus bradycardia, and atrioventricular conduction block. Cooperating patients underwent a cardiopulmonary exercise test with the assessment of peak oxygen uptake (VO2peak), respiratory exchange rate (RER), heart rate at maximum exercise, blood pressure response to exercise, and the presence of exercise-induced or exerted cardiac arrhythmias.

CMR was performed using a 1.5-T scanner (Sonata and Avanto fit, Siemens, Germany). Cine images were acquired by the breath-hold, electrocardiographic-gated, segmented k-space steady-state free-precession technique using 25 phases per cardiac cycle. LGE images were obtained in the long-axis and short-axis imaging planes using a breath-hold segmented inversion recovery sequence implemented 10–15 min after intravenous administration of 0.1 mmol/kg of gadobutrol (Gadovist, Bayer, Berlin, Germany); gadodiamide (Omniscan, GE Healthcare, United Kingdom) was used instead of gadobutrol if the patient was under 2 years of age. The criteria for the diagnosis of LVNC in the CMR study based on the assessment of the ratio of the NC/C layer of the LV myocardium being > 2.3 in the end-diastolic phase were used in accordance with the recommendations of the literature [10].

## 4. Results

**Baseline characteristics**. A total of 32 patients with echocardiographic diagnosis of LVNC were recruited between February 2008 and July 2020, with the median age being 11.5 (6–15) years, and the proportion of males being 53%. Patients were followed prospectively for a median of 4.02 (IQR 0.48–10.14) years. Clinical and demographic characteristics of the patient population are presented in Table 1.

**Table 1.** Patient characteristics in the whole left ventricular noncompaction cardiomyopathy (LVNC) cohort.

| Clinical Parameters | Total n = 32 |
|---|---|
| Age, cohort median, yrs (IQR) | 11.5 (6–15) |
| Age ≤1 yr, n (%) | 1 (3%) |
| Age >1 and ≤10 yrs, n (%) | 10 (31%) |
| Age >10 and <18 yrs, n (%) | 21 (66%) |
| **Positive family history, n (%)** | 17 (53%) |
| LVNC, n (%) | 10 (31%) |
| HCM, n (%) | 2 (6%) |
| DCM, n (%) | 2 (6%) |
| CHD, n (%) | 1 (3%) |
| Bradycardia, n (%) | 4 (13%) |
| WPW, n (%) | 2 (6%) |
| AVNRT, n (%) | 1 (3%) |
| SCD, n (%) | 3 (9%) |
| **Clinical symptoms, n (%)** | 11 (34%) |
| Chest pain, n (%) | 3 (9%) |
| Palpitations, n (%) | 2 (6%) |
| Syncope, n (%) | 3 (9%) |
| Pre-syncope, n (%) | 3 (9%) |
| Thromboembolic episodes, n (%) | 0 (0%) |
| NYHA functional class, **n (%)** | |
| I | 7 (22%) |
| II | 24 (77%) |
| III | 0 (0%) |
| IV | 1 (3%) |

**Table 1.** *Cont.*

| Clinical Parameters | Total n = 32 |
|---|---|
| Genetic syndrome, n (%) | 2 (6%) |
| **Increased NTproBNP value, n (%)** | 5 (16%) |
| NTproBNP value, median (IQR) | 349.40–27,057.00, 66.24 (25.71–105.35) |
| **Chest X-ray** | |
| CTR value, median (IQR) | 0.55–0.69, 0.56 (0.55–0.64) |
| Pulmonary congestion | 1 (3%) |
| **ECG, n (%)** | 32 (100%) |
| **ECG changes, n (%)** | 18 (56%) |
| Sinus bradycardia, n (%) | 7 (22%) |
| Nodal rhythm, n (%) | 2 (6%) |
| WPW, n (%) | 1 (3%) |
| RBBB, n (%) | 1 (3%) |
| LBBB, n (%) | 0 (0%) |
| LV overload, n (%) | 4 (13%) |
| ST-T changes, n (%) | 12 (38%) |
| Permanent cardiac pacing, n (%) | 1 (3%) |
| **Echocardiography, n (%)** | 32 (100%) |
| NC/C = 2.06–5.14, n (%) | 30 (94%) |
| NC/C = 1.9, n (%) | 1 (3%) |
| NC/C = 1.8, n (%) | 1 (3%) |
| Reduced LVEF acc. Simpson formula | 10 (31%) |
| LVEF 51–55%, n (%) | 7 (22%) |
| LVEF 46–50, n (%) | 2 (6%) |
| LVEF 40–45, n (%) | 1 (3%) |
| **CMR, n (%)** | 29 (91%) |
| NC/C = 2.3–6.24, n (%) | 24 (82%) |
| NC/C = 1.2–2.1, n (%) | 5 (17%) |
| **Pharmacological treatment, n (%)** | 22 (69%) |
| Beta-blockers, n (%) | 9 (28%) |
| ACE-I, n (%) | 19 (59%) |
| Furosemide, n (%) | 1 (3%) |
| Spironolaktone, n (%) | 16 (50%) |
| Acetylsalicylic acid, n (%) | 3 (9%) |
| Acenokumarol, n (%) | 1 (3%) |
| Salbutamol, n (%) | 4 (13%) |
| **Other procedures, n (%)** | 6 (19%) |
| Electrophysiological study, n (%) | 1 (3%) |
| RF ablation, n (%) | 1 (3%) |
| Pacemaker, n (%) | 2 (6%) |

**Table 1.** *Cont.*

| Clinical Parameters | Total n = 32 |
|---|---|
| LVAD, n (%) | 1 (3%) |
| W/L for HTx, n (%) | 1 (3%) |
| **Death, n (%)** | 1 (3%) |

HF—heart failure; SCD—sudden cardiac death; CTR—cardiothoracic ratio; NC/C—noncompaction to compaction layer ratio; EF—ejection fraction; RBBB—right bundle branch block; LBBB—left bundle branch block; EPS—electrophysiology study; ablation RF—radiofrequency ablation; LVAD—left ventricular assist device; LVNC—left ventricular noncompaction cardiomyopathy; HCM—hypertrophic cardiomyopathy; DCM—dilated cardiomyopathy; WPW—Wolff–Parkinson–White syndrome; AVNRT—atrioventricular nodal reentry tachycardia.

**Population characteristics.** In the study group, 3% of patients were under 1 year of age, 31% were between 1 and 10 years of age, while as many as 66% were over 10 years of age. In family history, as many as 17 (53%) children had cardiomyopathies in first-degree relatives (LVNC in 31% of children; HCM in 6% of patients, and DCM in 6% of patients). Family history of cardiac arrhythmias was observed in 7 (22%) patients, while sudden cardiac deaths occurred in the families of 3 (9.3%) children.

**Clinical presentation.** NYHA/Ross functional class evaluation demonstrated grade II in the majority of patients (77%); 3% had grade IV, while 22% of children had grade I. The most common presenting symptoms were syncope (9% of patients), pre-syncope (9%), chest pain (9%), and palpitations (6%). Two patients had evidence of dysmorphic features. One child was diagnosed with Barth syndrome and one with congenital microphthalmia. Serum NT-proBNP level was increased in five (16%) patients (from 349.40 to 27,057.00 pg/mL, with the reference value of up to 320 pg/mL).

**Electrocardiographic findings.** ECG abnormalities were noted in 18 (56%) children. The most prominent feature was ST-T changes, mainly T-wave inversion in 38% of patients, sinus bradycardia in 22%, and electrocardiographic features suggestive of LV overload in 13% of patients. Wolff–Parkinson–White syndrome (WPW) was noticed in 3% of patients (Figure 1).

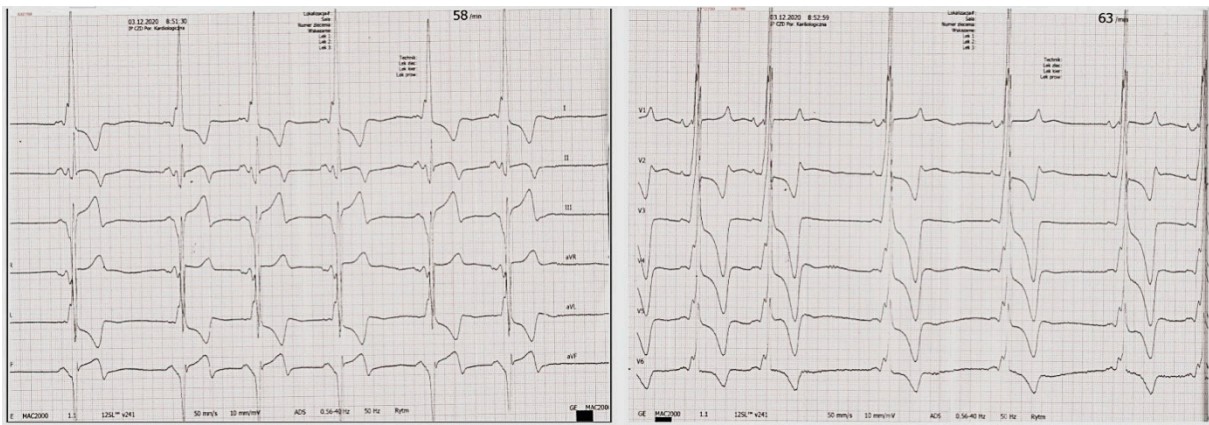

**Figure 1.** Twelve-lead ECG recording showing the features of the Wolff–Parkinson–White syndrome in a patient with LVNC.

In 24-h ECG Holter monitoring, the most prominent features were premature ventricular and atrial contractions in 25% and 16% of patients, respectively. Other findings, including sinus bradycardia, sick sinus syndrome, paroxysmal second- and third-degree atrioventricular block, and ventricular and supraventricular tachycardia, are presented in Table 2.

**Table 2.** Heart rhythm and conduction disturbances in the studied group of children.

| Heart Rhythm and Conduction Disturbances | Number of Patients, n = 32 (100%) |
|---|---|
| Supraventricular premature beats, n (%) | 5 (16%) |
| Ventricular premature beats, n (%) | 8 (25%) |
| Nodal rhythm, n (%) | 3 (9%) |
| Sinus bradycardia, n (%) | 7 (22%) |
| Sick sinus syndrome, n (%) | 2 (6%) |
| WPW syndrome, n (%) | 2 (6%) |
| Second degree paroxysmal block a-v, n (%) | 1 (3%) |
| Third degree paroxysmal block a-v, n (%) | 4 (13%) |
| VT, n (%) | 3 (9%) |
| Supraventricular tachycardia, n (%) | 1 (3%) |
| Atrioventricular tachycardia, n (%) | 1 (3%) |

WPW—Wolff–Parkinson–White syndrome; a-v—atrioventricular; VT—ventricular tachycardia.

**Two-dimensional echocardiography.** Most of the studied patients (94%) met the criteria for the diagnosis of LVNC. The LV myocardial NC/C ratio ranged from 2.06 to 5.14 (Figure 2). The use of color Doppler in the parasternal short-axis and apical four-chamber views improved the visualization of the trabeculations within the LV endocardium. LV function was reduced in 31% of patients (LVEF ranged from 40 to 55%).

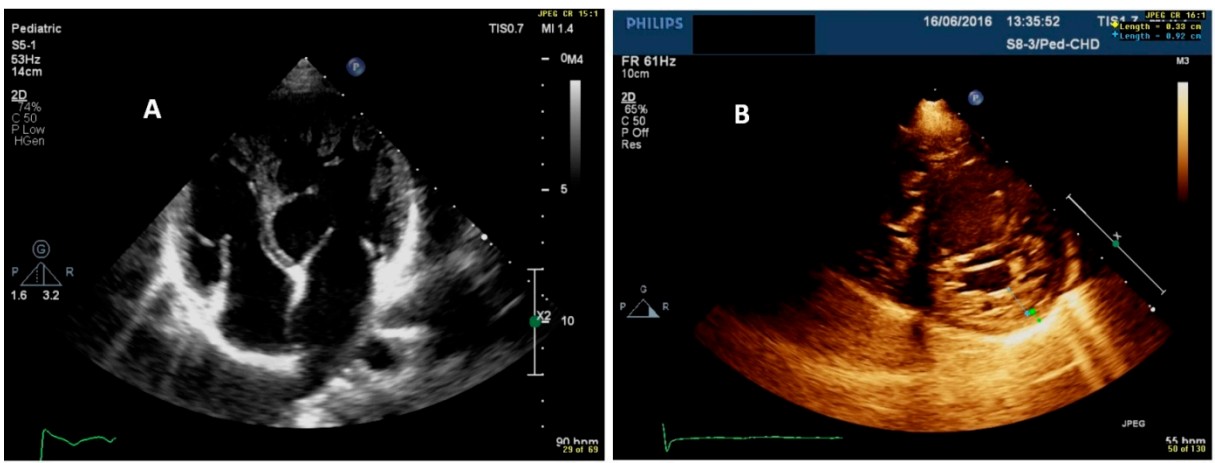

**Figure 2.** The 2D transthoracic echocardiography in a patient with LVNC. (**A**) Apical four-chamber projection: left ventricular noncompaction with deep recesses. (**B**) Parasternal short axis projection: the ratio of the left ventricular noncompacted to compacted layer (3:1).

**Cardiovascular magnetic resonance**. CMR was performed in 29 patients (91%); in one child, the parents did not consent to the study, and in two patients, a permanent pacemaker was implanted. CMR accurately delineated LV morphology in 24 (82%) patients, with the NC/C myocardial ratio in those children ranging from 2.3 to 6.2 (Figure 3).

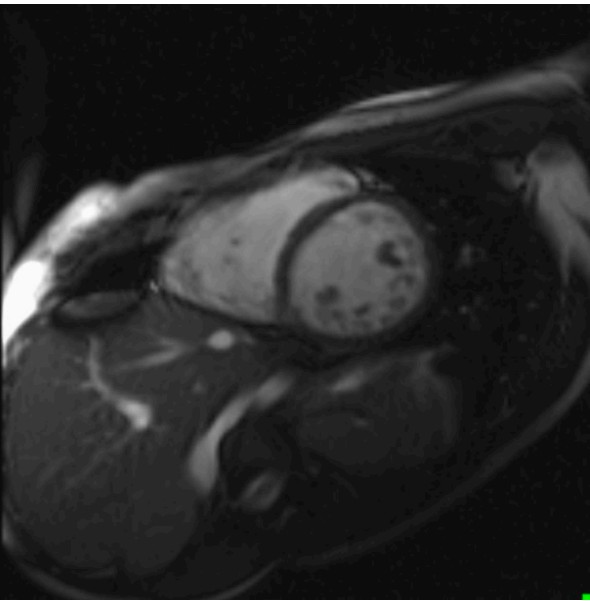

**Figure 3.** Cardiovascular magnetic resonance study in a patient with LVNC-short axis projection: noncompacted left ventricular layer.

**Cardiopulmonary exercise test (CPET)**. CPET (ergospirometry) was performed in 10 (31%) patients. Peak oxygen absorption (peak VO2) ranged from 15.9 mL/kg/min to 43.5 mL/kg/min, and in two patients (6%), it was below 18 mL/kg/min. The respiratory exchange rate (RER) ranged from 0.84 to 1.29 and was above 1.05 in six patients (19%). The heart rate at maximum effort ranged from 105/min to 195/min. All 10 patients had normal blood pressure responses to exercise. There were no exercise-induced arrhythmias in any of the patients.

**Genetics**. Pathogenic/likely pathogenic variants/rare variants of unknown significance in LVNC-associated genes were detected in 17 families (53%). Autosomal dominant defects were identified in the *HCN4, MYH7, RBM20*, and *TTN* genes. *ACTC1, ACTN2, DES, EYA4, HCCS, KCNQ1*, and *PRDM16* defects were found in single families. Syndromic LVNC (Barth syndrome and syndromic microphthalmia) was associated with *TAFAZZIN* and *HCCS* gene dysfunction in two patients, respectively. Complex genotypes (*KCNQ1* and *TTN*) were detected only once in our group of children. To date, segregation analysis was available in seven families, thus confirming the parental origin of identified variants in six cases and one de novo case.

**Medical therapy.** Treatment of congestive heart failure consisted of angiotensin-converting-enzyme inhibitors, spironolactone, beta-blockers, and diuretics in 19 patients (59%). Antithrombotic prophylaxis with acetylsalicylic acid was used in three (9%) patients with LVNC and significantly reduced LV systolic function. Salbutamol was used to treat sinus bradycardia in four patients (13%), and a pacemaker was implanted in two patients (6%), due to symptomatic sinus bradycardia and complete atrioventricular block, respectively. RF ablation was performed in one patient due to supraventricular tachycardia. In one child (3%), the LV mechanical support device (LVAD) was implanted; this patient died while being on the active list for heart transplantation.

## 5. Discussion

Today, due to advanced diagnostic tools, LVNC is a genetically determined cardiomyopathy that is being diagnosed with increasing frequency. The cause of the disease is thought to be a disruption of endomyocardial morphogenesis between the fifth and the eighth gestational week [12,13]. The variability in phenotypic expression and the lack of standard diagnostic criteria for the pediatric population make it difficult to estimate the frequency of this form of cardiomyopathy in children. In the opinion of many authors, the

presence of only morphological criteria characterized by a distinct two-layer structure of the LV muscle is insufficient for the diagnosis of LVNC, and a thorough clinical evaluation of the patient is also necessary [3,12,14]. In the literature, many studies on children with LVNC [3,15] concern a heterogeneous group of patients, including children with congenital heart disease. In our study, we present the clinical and diagnostic features of LVNC in children with an isolated form of this disease.

It should be emphasized that in the family history of our patients, as in the Ichida report [16], cardiomyopathies in family members occurred in 44% of children, including LVNC in as many as 31% of the studied children. Other authors also emphasize that the incidence of familial LVNC cases is higher in the pediatric population than in adults [5].

It should be noted that the main reason for referring children to a cardiology center was a family history of cardiomyopathy, sinus bradycardia, cardiac arrhythmias, and sudden cardiac deaths in family members. Of 32 patients, 13 (41%) children were referred for cardiac screening due to their family history. Cardiac arrhythmias were the first symptom of the disease in 10 (31%) children. In one (3%) infant, the initial presentation of the disease was symptoms of heart failure, which were the reason for echocardiography. The patient with Barth's syndrome was referred to perform, for the first time, cardiological diagnostics. In four (12.5%) children, postnatal echocardiography revealed left ventricular hypertrabeculation and these patients were referred to a cardiology center for comprehensive cardiological examinations. Heart murmur was the reason why three (9%) patients were referred for echocardiography with LVNC diagnosis.

It is noteworthy that sudden cardiac death occurred in the families of three children. In one child, two SCDs occurred in the family—in the father at 32 years of age with previously diagnosed HCM, and in his uncle at 42 (he was not examined before death, and no postmortem examination was performed). In the second family, SCD occurred in a 2-year-old sister of a studied patient, who had been diagnosed with LVNC before her death. In the third family, SCD occurred in the patient's aunt's one-month-old baby; no postmortem examination was performed.

Another interesting point is the association of LVNC with dysmorphic features. In our study, the genetic syndrome was diagnosed only in two children (6%), much less frequently than in the studies of the above-cited authors, where the frequency of coexistence of genetic syndromes was estimated at 14%, and even 37% and 66% [12,16,17]. In our group, one child was diagnosed with Barth syndrome with a confirmed mutation in the *TAFAZZIN* gene. The second patient had congenital microphthalmia and a pathogenic *HCCS* variant. It is extremely important that in our group of patients, as in the study by Dong X et al. [2], the molecular basis of LVNC was confirmed in 53% of cases. Corresponding with the existing literature [18], autosomal dominant inherited variants were more frequent in our cohort, with *HCN4* and *MYH7, RBM20*, and *TTN* defects accounting for 18% and 12% of the cases, respectively (data not shown). Segregation analyses confirmed the origin of the mutation from an affected parent.

The clinical course of LVNC in children is very diverse and may include symptoms of heart failure, cardiac arrhythmias, and thromboembolic events. According to the literature data, the most common symptom of LVNC in children is heart failure, the frequency of which is estimated at 35 to 91%, caused by LV systolic dysfunction [3,19–21]. Systolic dysfunction and a reduction in LVEF in patients with LVNC are associated with hypoperfusion secondary to subendocardial microvascular abnormalities and dyssynchronization between the integrated and noncompacted layers. Another possible explanation is dependence of the noncompacted area on aerobic oxidation and its sensitivity to hypoxia and the toxic effects of catecholamines [22]. In our group, symptoms of heart failure, including a reduction in the LVEF, were found in 31% of the studied children, with 16% of them having an increased serum NT-proBNP levels.

Most pediatric patients with LVNC had abnormalities in resting ECG, the most frequently reported changes being intraventricular conduction disturbances, including right bundle branch block, atrioventricular block, repolarization abnormalities, and LV over-

load features [3,22]. As in the literature reports [12,16,23], changes in the resting ECG occurred in 56% of our patients. Most often, that is, in as many as 38% of children, repolarization abnormalities in the form of ST-T changes were found, and 13% of patients had electrocardiographic features of LV overload. Literature reports indicate that severe sinus bradycardia and sinus node dysfunction are common complications in patients with LVNC [24,25]. Our observations confirm the above reports, as sinus bradycardia was diagnosed in as many as 22% of the studied children, and sick sinus syndrome in 6%. In one of them, with symptoms of sinus bradycardia, a permanent pacemaker was implanted. The fact of frequent short episodes of paroxysmal third-degree atrioventricular block (13% of patients) in our study deserves particular emphasis. One of those patients met indications for permanent pacemaker implantation. The results of published studies show a variable frequency of ventricular arrhythmias in patients with LVNC (from 6 to 60% of cases) [5,26]. It should be emphasized that in our group of patients, arrhythmias were a common symptom: as many as 25% of children had premature ventricular beats, 16% had premature supraventricular beats, 9% had short episodes of ventricular tachycardia, and 3% had supraventricular tachycardia (in this case, successfully treated with RF ablation). The literature describes cases of coexistence of LVNC and WPW syndrome. The authors emphasize that this relationship is more common in children than in adults, and its frequency was estimated at 8–14% [27]. In our study group, LVNC and WPW syndrome coexisted less often; cooccurrence was found in only 3% of patients. Brescia et al. [28] described a case of a patient with features of WPW syndrome on a resting ECG, but no accessory atrioventricular (AV) conduction pathway was found in electrophysiology studies (EPS). Similarly, in our study, one patient (3%) had electrocardiographic features of WPW syndrome on a resting ECG, while in the EPS examination, no true accessory AV conduction pathway was found. It should be noted that this patient had episodes of atrioventricular tachycardia, while echocardiography showed normal LV systolic function. In the second patient, a single episode of paroxysmal WPW syndrome was recorded in a 24-h Holter ECG, while this patient had a completely normal resting ECG recording, without WPW syndrome features. This patient was not tested for EPS. In paper by Howard TS et al. [29], they found, however, that LVNC and true WPW syndrome coexist in most cases, which worsens the prognosis in these patients. In the opinion of these authors, the presence of an accessory atrioventricular conduction pathway in patients with LVNC increases the risk of arrhythmias and sudden cardiac death, and also contributes to the development of left ventricular dyssynchrony, which may lead to a faster development of LV systolic dysfunction. It was also emphasized that RF ablation improved the systolic function of the LV [29,30], which further confirms the negative impact of the presence of WPW in patients with LVNC [29].

Systemic emboli are another important complication in patients with LVNC. Although their prevalence was as high as 38% two decades ago, a recent study reported it to be as low as 0–2%, and it was found to be 4–7% in another report [3,14]. There are no established recommendations for the use of antithrombotic prevention in children with LVNC, and the authors' opinions are divided [12,22]. None of our patients developed systemic emboli. Antithrombotic prophylaxis with aspirin was used in only four (13%) children with significantly reduced LVEF.

The first-line and standard procedure for diagnosing LVNC is 2-D Doppler echocardiography according to the criteria published in the literature [9,12,31,32]. CMR is increasingly used in the diagnostics of heart muscle diseases in the pediatric population [10,11,33,34]. It enables an accurate visualization of the heart muscle and a very reliable assessment of hemodynamic changes. It should be emphasized that, in our study, 94% of patients met the LVNC echocardiographic criteria, while the CMR study confirmed the diagnosis of the disease in 82% of children. In the remaining cases, in echocardiography and CMR, the ratio of the NC/C layer of the left ventricular muscle was borderline for the diagnosis of LVNC. Our previous research results confirm that there was also a good correlation of echocardiography with CMR in the group of patients with hypertrophic cardiomyopa-

thy [35]. The CPET is more and more frequently performed in the analysis of hemodynamic changes and in the assessment of exercise capacity in adult patients with cardiovascular diseases. It is still a rarely performed study in the pediatric population, especially with myocardial diseases. The literature shows that peak VO2 above 18 mL/kg/min is an important prognostic factor and is associated with a good 2-year prognosis, while peak VO2 below 10 mL/kg/min is associated with 36% annual mortality [36]. In our study, CEPT was performed in as little as 31% of patients, 6% of whom showed a low peak VO2, i.e., below 18 mL/kg/min. In the remaining 22 patients, CPET was not performed due to their age being below 10 years, the lack of appropriate equipment for this age group (n = 11 children), the psychological aspect and fear of wearing a face mask in 10 patients, and a significant spine defect in one child, which was a contraindication to the test.

Treatment of patients with LVNC should be directed towards three most important clinical manifestations: congestive heart failure, arrhythmias, and systemic embolic events. In our study, standard treatment of heart failure with preload and afterload reducers was started in all 31% of patients with systolic LV dysfunction and reduced LVEF. Moreover, patients with decreased LV systolic functions received a beta-blocker (bisoprolol), which has been shown to improve LV and neurohormonal dysfunction in children. Adwani et al. [37] described the case of the first successful heart transplantation in a patient with Barth syndrome. Since this report, heart transplantation has been recognized as an effective treatment for end-stage heart failure in patients with Barth syndrome. In our study, one patient diagnosed with Barth syndrome and severe heart failure (NYHA class IV) was implanted with a left ventricular assist device (LVAD) while awaiting a heart transplant. The patient died while on the waiting list for a heart transplant.

The results of the studies published in the literature, as well as the results of the authors' own research, indicate that LVNC is a myocardial disease with a varied clinical profile as well as a natural history that is not, at present, fully understood, which prompts the continuation of clinical trials involving higher numbers of patients.

## 6. What Are the Clinical Implications?

A diagnosis of LVNC in children is more likely in the context of a family history of cardiomyopathy.

The phenotype of familial LVNC in childhood is varied and includes severe cardiac symptoms, suggesting that clinical screening should commence at a younger age.

LVNC may be missed or overdiagnosed if echocardiography is the only imaging modality performed in a cardiac evaluation.

Genetic evaluation (testing and counseling) is recommended in each patient with isolated or syndromic LVNC.

Large-scale, multi-center, collaborative approaches to clinical and genetic evaluation in childhood LVNC are needed to develop robust standards of diagnostic and therapeutic management in this group of patients.

### *Limitations of the Study*

The limitations of this study are mainly inherited by its design. The study involved single-center research with a relatively small sample size. Needless to say, the results of this study need to be confirmed in large-scale, multicenter, longitudinal studies.

## 7. Conclusions

- Although heart failure and arrhythmias were very frequent in our study group, thromboembolic events and genetic syndromes were rare.
- Echocardiographic examination is the gold standard for the diagnosis of LVNC. However, cardiac CMR is recommended to confirm the diagnosis, especially in uncertain cases.
- Our results indicate that CMR has a good correlation with echocardiography and a high sensitivity and specificity in detecting non-compacted segments.

- For the accurate and reliable assessment of children with LVNC, it is necessary to get to know their family history and detailed clinical profile.
- The high genetic yield resulted in the explanation of molecular etiology in over half (53%) of the studied children.
- Identifying the genetic cause allows for risk stratification and may help in the clinical management and counseling of patients and their relatives.

**Author Contributions:** Conceptualization: L.Z., A.P. and D.P.-A.; Methodology: L.Z., A.P., A.M.-R., D.P.-A., E.C., Ł.M. and K.B.; Validation: L.Z., D.P.-A. and E.C.; Formal Analysis: L.Z., A.P., A.M.-R., D.P.-A. and E.C.; Investigation: A.P., L.Z., A.M.-R., D.P.-A., E.C., K.B. and Ł.M.; Resources: L.Z., A.P., D.P.-A. and Ł.M.; Data Curation, A.P., L.Z., D.P.-A. and E.C.; Writing—Original Draft Preparation, A.P., L.Z., D.P.-A., A.M.-R. and Ł.M.; Writing—Review and Editing, L.Z., A.P. and D.P.-A.; Supervision, L.Z. All authors have read and agreed to the published version of the manuscript.

**Funding:** This work was partially founded by The Children's Memorial Health Institute (statutory grant no. S177/2018).

**Institutional Review Board Statement:** The study was conducted according to the guidelines of the Declaration of Helsinki and approved by the Institutional Ethics Committee of The Children's Memorial Health Institute (protocol code 45/KBE/2018 and date of approval 24/Oct/2018).

**Informed Consent Statement:** Informed consent was obtained from all subjects and their parents involved in the study.

**Data Availability Statement:** The data presented in this study are available on request from the corresponding author.

**Conflicts of Interest:** The authors declare no conflict of interest. The funders had no role in the design of the study; in the collection, analyses, or interpretation of data; in the writing of the manuscript, or in the decision to publish the results.

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
