# Peer review of "Spectrum of Clinical Features and Genetic Profile of Left Ventricular Noncompaction Cardiomyopathy in Children"

_cardiogenetics, doi:10.3390/cardiogenetics11040020_

Round 1
Reviewer 1 Report
This paper summarizes the clinical and genetic findings in a series of children diagnosed with left-ventricular non-compaction in a single centre. It is a generally good study, which could, however, benefit from a few changes. Below, please find my comments.
Major:
The children were subjected to a genetic panel of 25 genes. Current panels, however, have significantly grown in size and scope. It has now become commonplace to evaluate up to 120 genes in a single test. If genetic material is still available, I would suggest that the children are subjected to broader testing.
It would significantly add to the study if family testing was performed on those children found to bear a genetic mutation.
The study would also benefit from the ‘initial presentation’; why were these children referred to the clinic for evaluation?
The study would finally benefit from two additional small sections; one at the introduction, stating ‘what is missing’; where is the gap in the knowledge that this paper is trying to address. The second in the end – stating what this paper is truly adding in terms of a clinical perspective. This should be highlighted more emphatically perhaps instead of listing the observations as bullet points.
Minor:
It is reported that sudden cardiac deaths occurred in the families of 3 children. Were these accompanied by a post-mortem diagnosis of a cardiomyopathy? This should also be mentioned.
Left-ventricular hypertrophy is noted under the ‘ECG findings’ section. It looks like an imaging finding – perhaps state that it refers to an ECG features suggestive of ventricular overload.
Why was CPET only performed in 10 children? Lack of consent from the families or phenotype of heart failure in this sub-group?
It is not clear why LVNC is more common in the paediatric population than in the adults. This notion would benefit from a short explanation/speculation.
There is no explanation or speculation on why thromboembolitic events were so rare in this series.
Reviewer 2 Report
Thank you for allowing me to review the study of Paszkowska et al. on LVNC in children. The work may be published only after some changes have been made.
The authors should discuss the observation of a 3% WPW pattern. This pattern can be mimicked by the LVNC phenotype. Please compare your findings to those of other authors.
Based on the gene discoveries, please discuss the following inheritance patterns: autosomal dominant, autosomal recessive, and X-linked recessive.
Please discuss the 17-genes Phosphorus left ventricular noncompaction panel with: Indications, Methodology, Performance characteristics and Interpretation.
Round 2
Reviewer 1 Report
Many thanks to the authors for addressing all the comments I raised in my original review. I have no further recommendations.